# MAGE-A3 is a prognostic biomarker for poor clinical outcome in cutaneous squamous cell carcinoma with perineural invasion via modulation of cell proliferation

**Aaron Chen**[1], **Alexis L. Santana**[1], **Nicole Doudican**[1], **Nazanin Roudiani**[1], **Kristian Laursen**[2], **Jean-Philippe Therrien**[3], **James Lee**[3], **Diane Felsen**[4], **John A. Carucci**[1]*

1 Ronald O. Perlman Department of Dermatology, New York University Langone Medical Center, New York, NY, United States of America, 2 Department of Pharmacology, Weill Cornell Medicine, New York, NY, United States of America, 3 GlaxoSmithKline, Research Triangle, NC, United States of America, 4 Pediatric Urology, Weill Cornell Medicine College, New York, NY, United States of America

* john.carucci@nyumc.org

**Data Availability Statement:** All relevant data are within the manuscript and its Supporting Information files.

## Abstract

Perineural invasion is a pathologic process of neoplastic dissemination along and invading into the nerves. Perineural invasion is associated with aggressive disease and a greater likelihood of poor outcomes. In this study, 3 of 9 patients with cutaneous squamous cell carcinoma and perineural invasion exhibited poor clinical outcomes. Tumors from these patients expressed high levels of MAGE-A3, a cancer testis antigen that may contribute to key processes of tumor development. In addition to perineural invasion, the tumors exhibited poor differentiation and deep invasion and were subsequently classified as Brigham and Women's Hospital tumor stage 3. Cyclin E, A and B mRNA levels were increased in these tumors compared with normal skin tissues ($102.93 \pm 15.03$ vs. $27.15 \pm 4.59$, $36.83 \pm 19.41$ vs. $11.59 \pm 5.83$, $343.77 \pm 86.49$ vs. $95.65 \pm 29.25$, respectively; $p < 0.05$). A431 cutaneous squamous cell carcinoma cells pretreated with MAGE-A3 antibody exhibited a decreased percentage S-phase cells ($14.13 \pm 2.8\%$ vs. $33.97 \pm 1.1\%$; $p < 0.05$) and reduced closure in scratch assays ($43.88 \pm 5.49\%$ vs. $61.17 \pm 3.97\%$; $p = 0.0058$). In a syngeneic animal model of squamous cell carcinoma, immunoblots revealed overexpression of MAGE-A3 and cyclin E, A, and B protein in tumors at 6 weeks. However, knockout of MAGE-A3 expression caused a reduction in tumor growth (mean tumor volume $155.3$ mm$^3$ vs. $3.2$ mm$^3$) compared with parental cells. These results suggest that MAGE-A3 is a key mediator in cancer progression. Moreover, elevated collagen XI and matrix metalloproteases 3, 10, 11, and 13 mRNA levels were observed in poorly differentiated cutaneous squamous cell carcinoma with perineural invasion compared with normal skin tissue ($1132.56 \pm 882.7$ vs. $107.62 \pm 183.62$, $1118.15 \pm 1109.49$ vs. $9.5 \pm 5$, $2603.87 \pm 2385.26$ vs. $5.29 \pm 3$, $957.95 \pm 627.14$ vs. $400.42 \pm 967.66$, $1149.13 \pm 832.18$ vs. $19.41 \pm 35.62$, respectively; $p < 0.05$). In summary, this study highlights the potential prognostic value of MAGE-A3 in clinical outcomes of cutaneous squamous cell carcinoma patients.

**Funding:** This study was funded in part by GlaxoSmithKline. A.S. was funded by CTSI TL1 TR001447 and research reported in this publication was supported in part by the National Institute of Arthritis and Musculoskeletal and Skin Diseases, part of the National Institutes of Health under Award Number T32AR064184. The content is solely the responsibility of the authors and does not necessarily represent the official views of the National Institutes of Health. The funders had no role in study design, data collection and analysis, decision to publish, or preparation of the manuscript.

**Competing interests:** Jean-Philippe Therrien and James Lee were employees and shareholders of GlaxoSmithKline while these studies were being performed. This does not alter our adherence to Plos One policies on sharing data and materials.

## Introduction

Cutaneous squamous cell carcinoma (cSCC) is the second most common human cancer responsible for approximately 10,000 deaths in the United States each year primarily due to complications from overwhelming tumor burden after nodal metastasis [1]. Perineural invasion (PNI) in cSCC is associated with aggressive disease and a greater likelihood of nodal metastasis with rates of 10% to 50% and subsequent disease-specific death. The reported incidence rates of PNI in cSCC range from 2.5% to 14% since most patients with cSCC and PNI present with no clinical symptoms and no radiologic evidence of PNI.

We previously demonstrated that expression of MAGE-A3, a cancer testis antigen (CTA), in cSCC is associated with advanced tumor stage and poor prognosis [2]. Cancer testis antigens (CTAs) are detected in many solid tumors as well as normal testis and placental tissues. CTAs contribute to key processes of tumor development, including stimulation of oncogenic pathways, such as cell proliferation, angiogenesis, and metastasis, and inhibition of tumor suppressor pathways [3]. Many studies have suggested that CTAs may represent valuable targets for the development of anti-cancer therapies with limited side effects [3–5]. Melanoma-associated genes (MAGEs) are CTAs expressed in various malignancies and have been widely studied as prognostic biomarkers [6–9]. Expression of the CTA MAGE-A3 correlates with aggressive clinical progression and drug resistance in variety of carcinomas, such as non-small cell lung carcinoma, diffuse large B-cell lymphoma, and gastric cancer [10–12]. MAGE-A3 expression is associated with enhanced cell proliferation and mediates fibronectin-controlled cancer progression and metastasis [12, 13]. Other factors, including cyclin proteins, may contribute to metastasis. Cyclin proteins partner with cyclin-dependent kinases (CDKs) to tightly control proliferation by regulating progression into G0/G1, S, G2 and M phases of the cell cycle. Given that altered cell cycle activity is commonly observed in cancer cells, regulatory proteins, such as cyclin D and E and CDKs, have been studied as biomarkers of cancer progression and targets of cancer therapy [14–18].

Herein, we studied a cohort of high risk cSCC patients and found that PNI cSCC was associated with increased expression of MAGE-A3 and cyclins E, A and B. We also found that elevated mRNA levels of collagen XI and matrix metalloproteases 3, 10, 11, and 13 were observed in poorly differentiated cutaneous squamous cell carcinoma with PNI.

## Materials and methods

All human studies were reviewed and approved by the institutional review board at NYU Langone Medical Center. Written informed consent was obtained for all patients before their participation, and the study was performed with strict adherence to the Declaration of Helsinki Principles. Human Subjects protocol: IRB protocol 16–00122.

Animal studies described were reviewed and approved by the Institutional Animal Care and Use Committee at NYU Langone Medical Center and were conducted according to the requirements established by the American Association for Accreditation of Laboratory Animal Care. All procedures were approved by the Institutional Animal Care and Use Committee before the initiation of any studies. Animal Protocol number: IA16-00510.

### Patients and samples

IRB approval at NYU Langone Medical Center was acquired prior to the study (IRB protocol 16–00122). Surgical discard that would otherwise be disposed of as medical waste was obtained from a total of 24 patients who had no history of organ transplants and presented for dermatologic evaluation of primary cSCC at NYU Langone Medical Center. Participation involved the use of surgical discard and correlation with clinical characteristics of the tumor. Participation

was voluntary. Among the participants, nine patients with primary cSCC who revealed PNI in their pathological reports were chosen for follow-up of their clinical outcomes, and their tumor specimens were analyzed for further investigation. Demographic information on these patients is reported in S1 Table. Other molecular characteristics of these patient tumors were described in another study [19].

All cSCC samples were selected from sun-exposed skin areas and obtained from debulking during Mohs micrographic surgery. Normal specimens were collected from non-sun-exposed areas of healthy volunteers with no history of cancers and organ transplants. Archived formalin-fixed paraffin-embedded (FFPE) blocks were acquired as previously reported in separate studies assessing potential biomarkers in SCC [2]. Four consecutive 15-μm sections were selected from each FFPE block using a standard microtome (Leica Instruments). The microtome blade was replaced, and equipment was sterilized using 100% isopropanol and Terminator RNase remover (Denville) between blocks.

## RNA extraction from FFPE

Total RNA was extracted from FFPE tissue samples using the Agencourt FormaPure kit (Beckman Coulter) per the manufacturer's protocol. The final samples were suspended in 80 μl RNase/DNase-free water, and purity was measured using a NanoDrop Reader (NanoDrop 2000C, Thermo Fisher Scientific).

## Nanostring analysis

Total RNA quality was assessed using the Agilent 2200 TapeStation Bioanalyzer. Total RNA samples were processed using a custom probe set via the nCounter Analysis System (Nano-String Technologies) per the company's protocol. The raw quantification that resulted from the nCounter System's barcode analysis was normalized using the nSolver software (NanoString Technologies) with relation to housekeeping genes. Data obtained via the NanoString nCounter system were analyzed using NanoString nSolver software. MAGEA3 positivity is defined by a normalized NanoString value of greater than 20 as described in our previous study [2].

## Cell lines and cell culture

A431 cells were obtained from American Type Culture Collection (ATCC CRL-1555; Manassas, VA) and grown in Dulbecco's modified Eagle's medium supplemented (DMEM) with 10% fetal bovine serum (FBS) (Gibco; Waltham, MA). All Pam 212 cell lines, including Pam 212 (gift from Dr. Stuart Yuspa, NIH, National Cancer Institute) [20], Pam 212 LY2 (gift from Dr. Carter Van Waes, National Institute on Deafness and Other Communication Disorders) [21], and the Pam 212 altered by CRISPR/Cas 9 were grown in DMEM with 10% FBS supplemented with 2 mM L-glutamine and 1 mM sodium pyruvate (Gibco), were grown in an incubator at 37˚C for 48 hours before experiments. For cell collection, cells were washed once using 1X phosphate buffer saline (PBS; Gibco) and digested using 0.25% trypsin (Gibco), which was terminated following addition of serum containing media. All cell cultures were grown at 37˚C in 5% $CO_2$. Prior to use in experiments, all cells were routinely assessed for mycoplasma contamination using the MycoAlert Mycoplasma Detection Kit (Cat # LT07-418, Lonza).

## Mice

All animals were handled in strict accordance with good animal practice as defined by the National Institutes of Health (NIH) Guide for the Care and Use of Laboratory Animals, and

the Public Health Service (PHS) Policy on Humane Care and Use of Laboratory Animals. Six-to eight-week-old female BALB/c mice were purchased from Harlan Laboratories (Indianapolis) and maintained at the NYU Langone Skirball animal facility. Mice were maintained in ventilated cages and fed/watered ad libitum with experiments performed under an IACUC approved protocol (160103–01) as well as following institutional guidelines for the proper and humane use of animals in research. For tumor growth experiments, 5.0 x $10^6$ Pam 212 or Pam 212 guide RNA (520 or 521) cells were injected intradermally with Matrigel under anesthesia to minimize pain and discomfort. After tumor implantation, mice were monitored thrice weekly for signs of pain or distress. Mice meeting any of the following criteria were immediately euthanized by $CO_2$ euthanasia: ulcerated tumors, greater that 20% of weight loss, signs of difficulty breathing or appear moribund, difficulty moving freely in the cage, or tumor burden greater than 15% body weight. Tumor growth measured weekly by the caliper method. Tumor volume was calculated using the formula a $2 \times$b/2, in which a is the short diameter of the tumor and b is the long diameter of the tumor. At the 6th week, mice were sacrificed by $CO_2$ euthanasia, and tumors were extracted. All experiments were conducted as approved by the Institutional Animal Care and Use Committee (Study number IA16-00510).

## Plasmids

To generate an inducible Cas9 construct for MAGE-A3 knockout, we employed the pBig-TRE3Gr eCas9 vector to generate constructs which express MAGE-A3 gRNAs (U6-driven) together with a dox-inducible TRE3G EGFP-IRES-eCas9 cassette. Embedded in the same construct is a PGK Neomycin-IRES-rtTA3 cassette, which provides a 3[rd] generation reverse-Tetracycline Activator (rtTA3). To target MAGE-A3, ds-oligos harboring gRNA sequences (-3, -4, and -5) were cloned into BbsI opened eSpCas9 (Addgene #71814). MAGE-A3 gRNA expression cassettes were PCR-amplified with the CRISPR(-)XbaI (5'–GGTACCTCTAGAGCCAT TTGTCTGC-3')/hU6(+)NheI (5'– tttgctagcGAGGGCCTATTTCCCATGAT -3') primer pair, and the purified NheI/XbaI fragment was cloned into either XbaI opened pBig-TRE3Gr (MAGE-A3-5) or XbaI opened pBigTRE3Gr MAGE-A3-5 (MAGE-A3-3 and -4). This procedure generated constructs pBigTRE3Gr MAGE-A3-5/3 (guide RNA 520) and pBig-TRE3Gr MAGE-A3-5/4 (guide RNA 521) with gRNAs for dual (1280) and (366) targeting (the numbers in parenthesis indicate the bp distance between the dual Cas9 target sites). The dual (366) targeting removes the splice-acceptor site of the coding exon, whereas the dual (1280) targeting removes the entire coding region of MAGE-A3 in a dox-dependent manner. The sequences of the targeting vectors are listed in Table 1.

## Generation of cell lines

Pam 212 cells were seeded at a density of 8.0 x $10^5$ cells per well into 24 well plates. After 24 hours, cells were transfected with CRISPR vectors 520 or 521 using lipofectamine 2000 reagent

**Table 1. Sequences of the targeting vectors.**

|  | Vector sequence |
|---|---|
| **Cas9MAGE-A3(-)5** | 5'-aaacCTATCCTTTCTCCATCAGGCC-3' |
| **Cas9MAGE-A3(+)5** | 5'-caccgGCCTGATGGAGAAAGGATAG-3' |
| **Cas9MAGE-A3(-)4** | 5'– aaacACTTCATTTGTTGCACAATGC-3' |
| **Cas9MAGE-A3(+)4** | 5'– caccgCATTGTGCAACAAATGAAGT-3' |
| **Cas9MAGE-A3(-)3** | 5'– aaacGCTTTGCTGAATGTCATCATC-3' |
| **Cas9MAGE-A3(+)3** | 5'– caccgATGATGACATTCAGCAAAGC-3' |

(Carlsbad, CA) according to the manufacturer's protocol. Doxycycline (2 ug/mL) was added 24–48 hours post-transfection and the GFP positive cells were sorted using a BD Biosciences FACSAria Fusion flow cytometer. Then, 96-well serial dilution was performed using expanded GFP+ sorted cells followed by screening for MAGE-A3 knockout.

## PCR validation of Pam 212 CRISPR cell lines

Complete genomic DNA of Pam 212 wild type, Pam 212 with guide RNA 520 (Guide 520) and Pam 212 with guide RNA 521 (Guide 521) was isolated separately by the Qiagen DNeasy Blood and Tissue kit (QIAGEN, Cat No. 69506) according to the manufacturer's protocol. Afterwards, PCR was performed using Platinum Taq High Fidelity polymerase (Invitrogen, Carlsbad, CA) in a volume of 25 μl containing 1X Platinum Taq High Fidelity Buffer (MgCl$_2$-free), 0.2 mM dNTPs, 1 U Ex Taq DNA polymerase, and 5 mol each of the aforementioned forward and reverse primers. A total of 500 ng extracted genomic DNA was added to the reaction mixture. The following thermal cycling profile was utilized: 98˚C for 30 sec followed by 30 cycles of 98˚C for 10 sec, 60˚C for 90 sec and 72˚C for 2 min. The last cycle was followed by a final extension step of 5 min at 72˚C.

## EDU assays

For EDU assays, A431 cells were seeded at 1.0 x 10$^5$ cells per well in 6-well plates and treated with MAGE-A3 antibody at a 1:500 dilution mixed with antibody transport solution (ATS), which contained PBS, 50% glycerol, 0.02% sodium azide PH 7.3 and 0.1% dimethylsulfoxide (DMSO). At 72 hours post-treatment, the Click-IT EDU assay was performed according to the manufacturer's protocol (Invitrogen, Carlsbad, CA). Flow cytometry was performed using BD Biosciences LSR II (BD Biosciences, San Jose, CA), and data were analyzed using FloJo2 software (BD Biosciences, San Jose, CA).

## Scratch assays

A431 cells were seeded into 24-well plates at a density of 5 x 10$^5$ per well and grown to 90% confluence. A scratch was made in the cell monolayer using a 1-mL pipette tip 24 hours after incubation with 5% CO$_2$ at 37˚C. Afterwards, the cells were washed with warmed 1 X PBS and media replaced with or without MAGE-A3 antibody treatment, the mixture of MAGE-A3 antibody and ATS at 1:500 dilution as described above. Scratch closure was photographed at 0, 24, 48, and 72 hours. The percentage of closure was calculated using ImageJ software.

## Immunoblot analysis and antibodies

For immunoblot analysis, cells were lysed in RIPA and whole cell lysates resolved by 10% SDS-PAGE and transferred to PVDF membrane. Immunoblots were probed using the following antibodies where specified: MAGE-A3 IgG1-kappa monoclonal antibody (Proteintech 60054-1-Ig), MAGE-A4 (Cell Signaling (E7O1U) XP Rabbit mAb #82491), p53 (Cell Signaling (1C12) Mouse mAb #2524), Actin IgG1-kappa monoclonal antibody (Thermo Fisher #MA5-11869), GAPDH (Cell Signaling 14C10) and cyclin antibodies (Cell Signaling: Cyclin A2 (BF683), Cyclin B1 (4138), Cyclin D1 (92G2), Cyclin D2 (D52F9), Cyclin D3 (DCS22), and Cyclin E1 (HE12)). Anti-mouse (Cell Signaling 7076s) and anti-rabbit (Cell Signaling 7074s) HRP-conjugated secondary antibodies were used for detection using HRP detection reagent (Thermo Scientific product # 1859698) and autoradiography film (Denville Scientific cat # E3012).

**Table 2. Primers for qRT-PCR.**

|  | Forward | Reverse |
|---|---|---|
| **Murine MAGE-A3** | CAGAGCCTACCCTGAAAAGTATG | AGCATCTGTTCAAGATCCAGGT |
| **Murine MAGE-A4** | GTCTCTGGCATTGGCATGATAG | GCTTACTCTGAACATCAGTCAGC |
| **Murine GAPDH** | AGGTCGGTGTGAACGGATTTG | GGGGTCGTTGATGGCAACA |
| **Murine CCND1** | GCGTACCCTGACACCAATCTC | ACTTGAAGTAAGATACGGAGGGC |

All gene sequences are listed in 5' to 3' order.

## Immunohistochemistry

Briefly, FFPE sections were incubated for 1 hour at 60°C followed by online deparaffinization. Antigen retrieval was performed using protease-3 (Ventana Medical Systems) for 12 minutes. Endogenous peroxidase activity was blocked with hydrogen peroxide. Samples were incubated in indicated antibodies for 1 hour at 37°C followed by biotinylated, goat anti-rabbit (Vector Laboratories Cat# Ba-1000 Lot# ZA0324 RRID: AB_2313606) diluted 1:200 in Tris-BSA and incubated for 30 minutes at 37°C. This step was followed by the application streptavidin-horseradish-peroxidase conjugate. Antibodies were visualized with 3,3 diaminobenzidene and enhanced with copper sulfate. Slides were washed in distilled water, dehydrated and mounted with permanent media. Appropriate positive and negative controls were included with study samples.

## Quantitative RT-PCR

For qRT-PCR analysis, cells were lysed in RTL buffer (Qiagen) supplemented with 1% BME prior to total RNA column purification (Qiagen RNEasy Mini Kit). CDNA generation and real-time PCR reactions were performed using the SuperScript III Platinum One-Step Quantitative RT-PCR kit with Rox by Invitrogen (Carlsbad, California). For MAGE-A3 transcript validation in CRISPR cell lines, Pam 212, Guide 520 and Guide 521 cells were lysed in RTL buffer (Qiagen) supplemented with 1% BME prior to total RNA column purification (Qiagen RNEasy Mini Kit) and DNase treatment (Ambion Turbo DNA Free kit) according to the manufacturers' instructions. DNase-free RNA was reverse transcribed using SuperScript III (Invitrogen), and cDNA was analyzed by qPCR using Taqman Gene expression assay with primers specific for murine MAGE-A3, MAGE-A4, Cyclin D1, and GAPDH (Table 2). RT-PCR was performed using the Applied Biosystems Step-one Real Time PCR system (Foster City, CA).

## Statistical analysis

Fischer's exact test was used to compare mRNA expression of genes of interest among different groups in the analysis of NanoString results. Statistical analysis of all RT-PCR and proliferation assay data was performed using GraphPad Prism version 7 (GraphPad Software, La Jolla, USA). One-way ANOVA followed by Tukey's multiple comparison correction was performed. For all tests, $p < 0.05$ was considered statistically significant.

## Results

### MAGE-A3 and cell cycle-associated cyclins are upregulated in human cSCC tumors exhibiting PNI

Tissue samples from 16 patients, including seven normal patients and nine patients with cSCC and PNI, were subject to RNA analysis using the Nanostring platform. MAGE-A3, MAGE-A4, cyclin D, cyclin E, cyclin A and cyclin B mRNA expression levels in tumor samples categorized

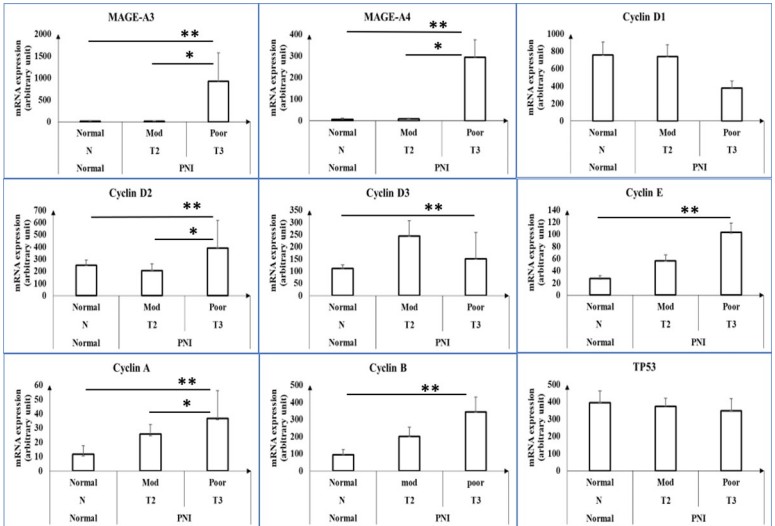

**Fig 1. Expression of MAGE-A3, MAGE-A4 and cyclins mRNA in normal skin and cSCC tumors with PNI.**
Expression levels of MAGE-A3, MAGE-A4, cyclin D1-3, cyclin E, cyclin A, cyclin B and p53 mRNA in normal skin tissues (n = 7), cSCC tumors with PNI and moderate differentiation (n = 6), and cSCC tumors with PNI and poor differentiation (n = 3) as assessed by Nanostring analysis. Group names are listed on the x-axis based on BWH tumor stage and tumor differentiation. Mod = moderate differentiation; Poor = poor differentiation; N = normal; T2 and T3 = BWH tumor stage. * and ** represent p<0.05.

by Brigham and Women's Hospital Tumor Staging System and tumor differentiation are shown in Fig 1.

Poorly differentiated cSCC with PNI exhibits the highest level of MAGE-A3 expression compared with moderately differentiated cSCC with PNI (924.74±653.82 vs. 16.62±6.815; p<0.0001). Similar expression trends are noted for MAGE-A4, but MAGE-A3 levels were significantly increased in poorly differentiated cSCC with PNI compared with MAGE-A4 (924.74 ±653.82 vs. 293.71±79.62; p = 0.0146). TP53 expression levels were similar in normal skin and BWH T2 and T3 cSCC tumor samples. Immunohistochemistry further confirms increased MAGE-A3 and MAGE-A4 expression in cSCC with PNI with poor differentiation compared with moderate differentiation (Fig 2).

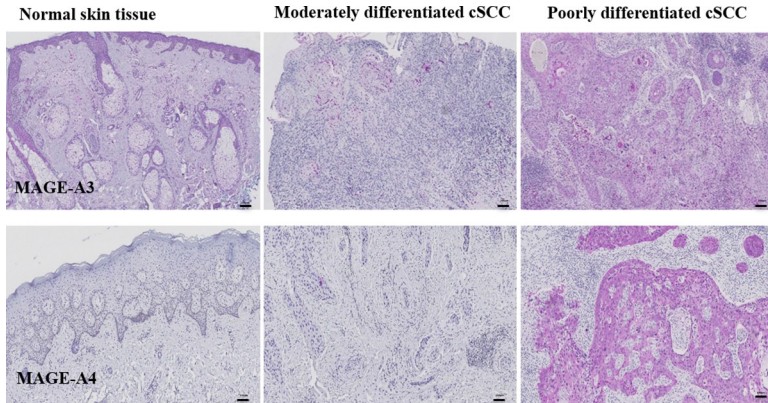

**Fig 2. Immunohistochemistry of MAGE-A3 and MAGE-A4 expression in normal skin tissue and moderately and poorly differentiated cSCC tumors.** Images in the upper row represent MAGE-A3 expression as assessed by immunohistochemistry, whereas images in the bottom row depict MAGE-A4 expression. Upregulated expression of MAGE-A3 and MAGE-A4 is noted in poorly differentiated cSCC with PNI. Scale bar in each image indicates 100 μm.

Given that cSCC tumors with PNI are associated with enhanced proliferation as evidenced by tumor size > 2 cm at presentation, we assessed the levels of cell cycle-related genes in these tumors [22]. As shown in Fig 1, poorly differentiated cSCC with PNI exhibits no statistically significant difference in cyclin D1 levels compared with other groups, including normal epithelial tissues ($p > 0.05$). In contrast, cyclin A, E, and B are expressed at higher levels in poorly differentiated cSCC with PNI compared with normal tissues ($p = 0.0097$, $p = 0.01$, and $p = 0.0167$, respectively). However, cyclin D2 and D3 levels in poorly differentiated cSCC with PNI were significantly increased compared with normal tissues ($391.35 \pm 227.5$ vs. $247.85 \pm 43.33$ ($p = 0.000945$) and $149.66 \pm 109.46$ vs. $111.35 \pm 14.69$ ($p = 0.000134$), respectively). Cyclin A is the only cyclin that is significantly increased in cSCC tumors with PNI with moderate differentiation compared with poor differentiation ($36.83 \pm 19.41$ vs. $25.68 \pm 6.86$; $p = 0.0028$).

## Blocking MAGE-A3 modulates cell cycle progression in A431 cells

We evaluated the role of MAGE-A3 in cell cycle progression using a 5-ethynyl-2′-deoxyuridine (EDU) incorporation assay. A431 cSCC cells were treated with anti-MAGE-A3 antibody vs. control, and the effect on the percentage of S-phase cells was assessed (Fig 3A). Reduced levels of S-phase A431 cSCC cells were observed upon treatment with MAGE-A3 antibody compared with antibody transport solution (ATS) alone and untreated groups ($14.13 \pm 2.8\%$ vs. $24.33 \pm 3.19\%$ vs. $33.97 \pm 1.1\%$, respectively; $p < 0.05$). Moreover, immunoblot analysis reveals increased p53 expression but no change in cyclin D1 expression in A431 cSCC cells pre-treated with MAGE-A3 antibody compared to controls (Fig 3B). We next assessed the impact of

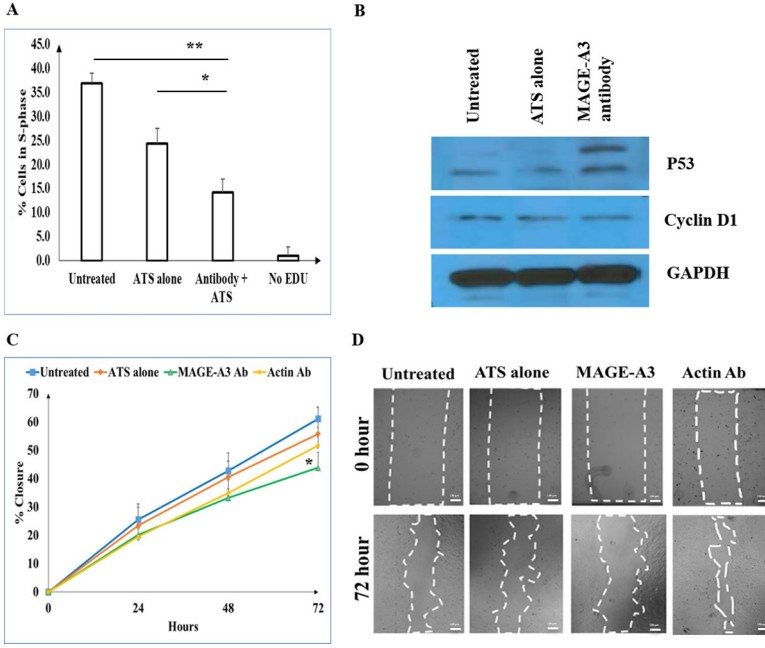

**Fig 3. MAGE-A3 antibody treatment reduces the number of S-phase A431 cells, impedes cell migration in scratch assays, and increases p53 expression by interacting with MAGE-A3 proteins.** (A) Percentage of A431 cells in S-phase in response to the following treatments: untreated, ATS alone, MAGE-A3 antibody + ATS, and untreated without EDU as a negative control. (B) Immunoblot of p53 and cyclin D1 expression in A431 cells subject to the following treatments: untreated, ATS alone and MAGE-A3 antibody. (C) Percentage closure in the different treatment groups over 72 hours. Actin antibody is used as an isotype control. The percentage closure in the group pre-treated with MAGE-A3 antibody was reduced after 72 hours compared with the untreated group ($43.88 \pm 5.49\%$ vs. $61.17 \pm 3.97\%$ ($p = 0.0058$), respectively). (D) Light microscopy images of scratch closures between 0 and 72 hours in the noted treatment groups. Dotted lines highlight the wounded area. Scale bars indicate 100 μm. * and ** represent $p < 0.05$.

MAGE-A3 antibody treatment on A431 cell migration using scratch assays. The percentage closure in the group pre-treated with MAGE-A3 antibody vs. control was reduced after 72 hours compared with the untreated group (43.88±5.49% vs. 61.17±3.97% (p = 0.0058), respectively; Fig 3C and 3D).

## Increased MAGE-A3 expression in an in vivo model of cSCC is associated with altered cyclin expression

Intradermal injection of Pam 212 cSCC cells into five syngeneic BALB/c mice resulted in locally invasive cSCC growth over six weeks (Fig 4A). Tumors exhibited significantly increased volume at week 6 compared with week 1 (1112.24± 34.6 mm$^3$ vs. 29.7± 10.9 mm$^3$; p = 0.02). MAGE-A3 and cyclin A2, B1 and E1 protein levels were increased in Pam 212 cSCC tumors compared with normal keratinocytes and Pam 212 cSCC cells grown in vitro, whereas cyclin D1 levels were reduced (Fig 4B). No significant differences in Cyclin D2 and D3 protein expression were noted as assessed by Western blot. In Fig 4C, significantly elevated MAGE-A3 and MAGE-A4 RNA transcript levels were also detected by qPCR in primary Pam 212 cSCC tumors compared to normal murine keratinocytes and Pam 212 cells in culture (77.4± 46.2 vs. 1± 0.1 vs. 2.62± 2.1; 90.4± 68.3 vs. 1± 0.1 vs. 1.86± 1.19, respectively; all p < 0.05). However, cyclin D1 levels were not significantly different (3.41± 0.12 vs. 1± 0.07 vs. 4.46± 0.32, p>0.05).

## Knockout of MAGE-A3 expression reduces cSCC tumor growth

Knockout of MAGE-A3 expression in Guide 520 and 521 was confirmed by PCR, qPCR and Western blot (Fig 5A–5C). These MAGE-A3 knockout cells along with wild type Pam 212 cells

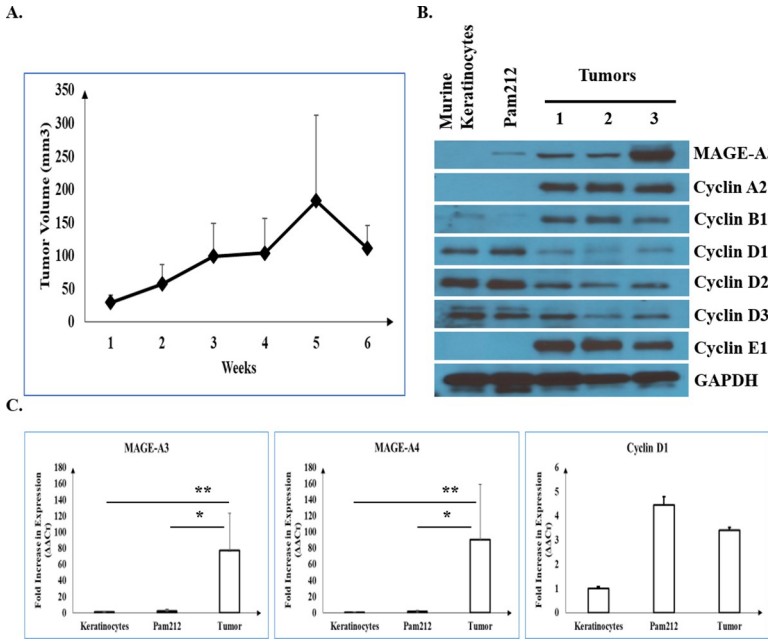

**Fig 4. Pam 212 cells form cSCC tumors in Balb/c mice with concomitant increases in MAGE-A3 and cyclin expression.** (A) Pam 212 cell growth in a syngeneic tumor model in 5 Balb/c mice over 6 weeks. (B) Western blot of MAGE-A3, cyclin A2, B1, D1-3 and E1 in normal murine keratinocytes, Pam 212 cells grown in culture, and Pam 212 cells grown as tumors in Balb/c mice. (C) Fold increase in MAGE-A3, MAGE-A4 and cyclin D1 mRNA expression in in normal murine keratinocytes, Pam 212 cells grown in culture, and Pam 212 cells grown as tumors in Balb/c mice as determined by qPCR. Fold increase of expression in normal murine keratinocytes was set to 1 ± 0. * and ** represent p<0.05.

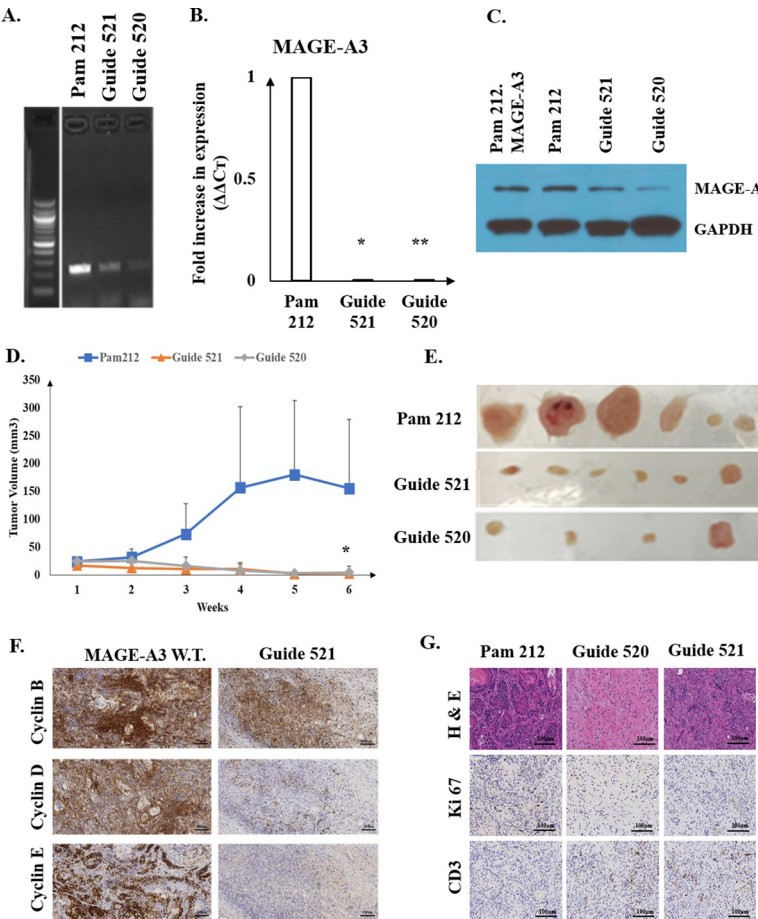

**Fig 5. Knockout of MAGE-A3 expression in Pam 212 by CRISPR guide RNA 521 and 520 results in reduction of syngeneic tumor growth in Balb/c mice.** (A) Validation of MAGE-A3 gene disruption was assessed by PCR. (B) RNA knockout was confirmed by qRT-PCR. Fold increase of MAGE-A3 expression in wild type Pam 212 was set to $1 \pm 0$. Fold changes in MAGE-A3 expression in Guide 521 and 520 were $0.0049 \pm 0.00198$ and $0.000535 \pm 0.000209$, respectively. Error bars indicate standard deviations. (C) MAGE-A3 suppression in Guide 521 and 520 was confirmed by Western Blot. Pam 212.MAGE-A3 represents Pam 212.pLKO.3G.MAGE-A3 as a positive control. (D) Volumes changes of Pam 212, Guide 521, and Guide 520 tumors in Balb/c mice over 6 weeks (n = 3 per group). The experiment was conducted twice with a total of 18 mice. At the 6th week, the volumes of syngeneic tumors of Guide 520 and 521 were reduced to $4.8 \pm 11.2$ mm$^3$ and $3.2 \pm 5.3$ mm$^3$ compared to Pam 212 tumors ($155.3 \pm 124.3$ mm$^3$). (E) Images of tumors obtained from mice after 6 weeks of growth in Balb/c mice. NOTE: Guide 520 yielded total tumor regression in 2 mice, so these tumors are not displayed. (F) Immunohistochemistry analysis of cyclin B, cyclin D, and cyclin E expression in MAGE-A3 wild type (W.T.) and Guide 521 tumor tissue. Scale bars indicate 100 μm. (G) Immunohistochemical assessment of MAGEA3, Ki67, and CD3 expression in Pam 212, Guide 520 and Guide 521 cSCC tumor tissue. Scale bars indicate 100 μm. * and ** represent p<0.05.

were injected intradermally into BALB/c mice, and tumor growth was assessed over a 6-week period. The results showed that MAGE-A3 knockout reduced the growth of Pam 212 cells in a syngeneic model of SCC (Fig 5D and 5E). At the 6th week, the volumes of syngeneic tumors of Guide 520 and 521 were reduced to $4.8 \pm 11.2$ mm$^3$ and $3.2 \pm 5.3$ mm$^3$ compared to tumors generated from parental cells ($155.3 \pm 124.3$ mm$^3$). These tumors also exhibited reduced levels of cyclin B, D, and E (Fig 5F). Moreover, reduced proliferation in these tumors was also supported by reduced Ki67 staining (Fig 5G).

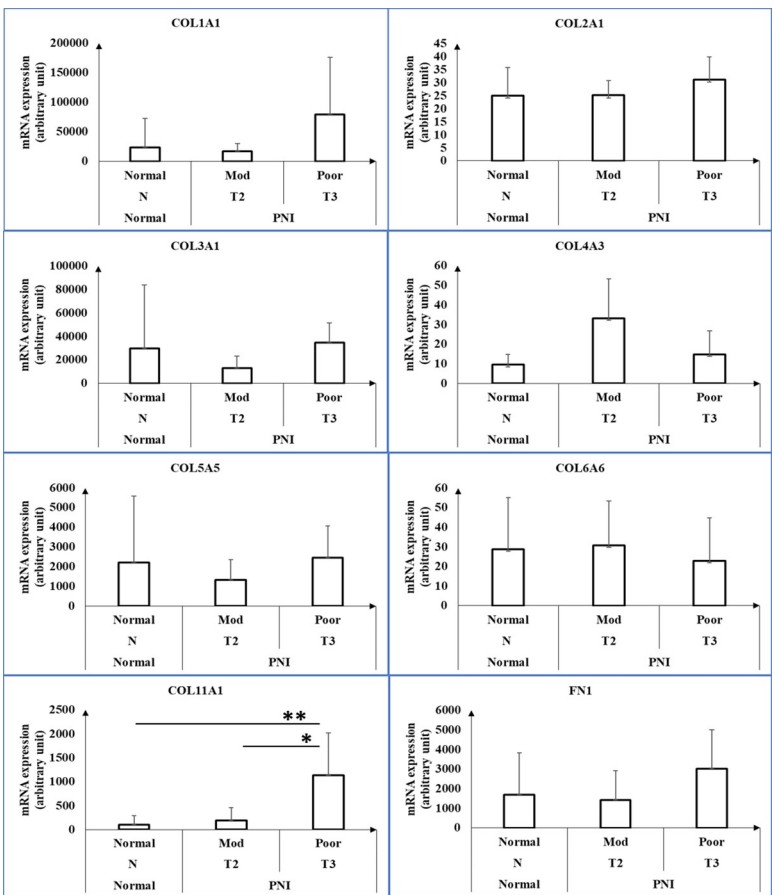

**Fig 6. Expression of collagens in normal skin and cSCC tumors with PNI.** Collagen and fibronectin mRNA levels in normal skin (n = 7), cSCC tumors with PNI and moderate differentiation (n = 6), and cSCC tumors with PNI and poor differentiation (n = 3) as assessed by Nanostring. Three groups were listed on x-axis based on BWH tumor stage and tumor differentiation. Mod = moderate differentiation; Poor = poor differentiation; N = normal; T2, T3 = BWH tumor stage. * and ** represent p<0.05. COL1A1 = collagen I alpha 1 chain; COL2A1 = collagen II alpha 1 chain; COL3A1 = collagen III alpha 1 chain; COL4A3 = collagen IV alpha 3 chain; COL5A5 = collagen V alpha 5 chain; COL6A6 = collagen VI alpha 6 chain; COL11A1 = collagen XI alpha 1 chain; FN1 = fibronectin.

## Collagen and matrix metalloprotease genes are upregulated in human cSCC tumors exhibiting PNI

We also profiled the expression of collagens (Fig 6) and matrix metalloproteases (MMPs) of patients with primary cSCC and PNI (Fig 7) to assess the interaction between carcinoma and extracellular matrix (ECM). Collagen XI was the only collagen elevated in poorly differentiated PNI compared with other two groups (1132.56±882.7 vs. 197.6±261.24 vs. 107.62±183.62; p< 0.05). MMP3, 10, 11 and 13 expression was significantly increased in poorly differentiated cSCC tumors with PNI compared with moderately differentiated (p<0.05). However, MMP11 levels were not different compared with poorly differentiated cSCC tumors with PNI and normal skin tissue (p = 0.325).

## Discussion

Numerous studies have demonstrated that SCC with PNI is correlated with poor clinical outcome and may benefit from adjuvant treatment post operatively due to increased incidence of

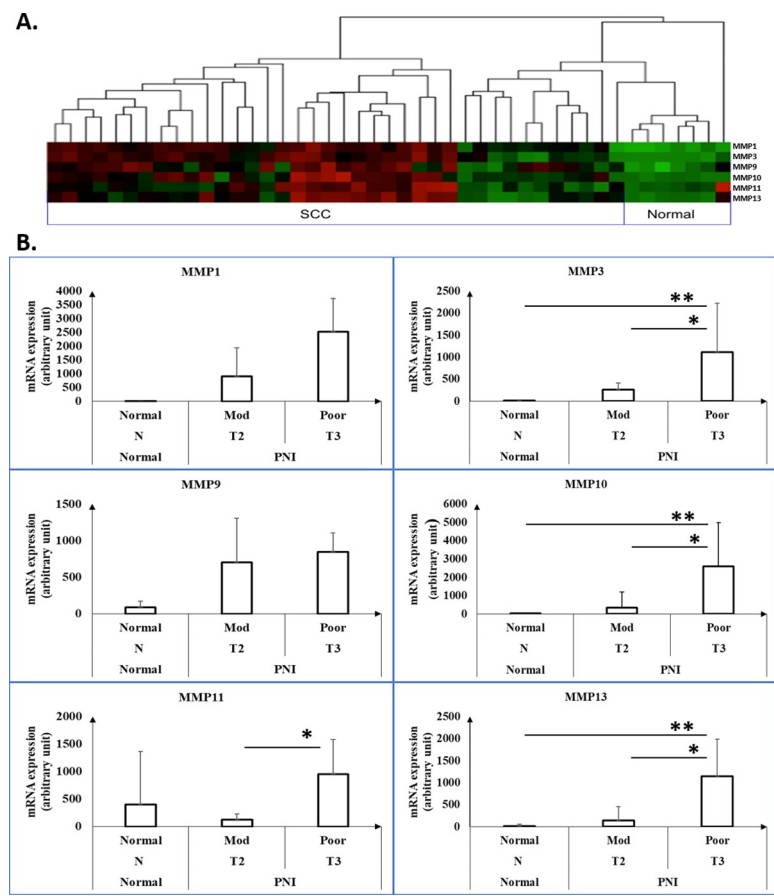

**Fig 7. Expression of MMPs in normal skin and cSCC tumors with PNI based on BWH tumor stage and tumor differentiation.** (A) The nSolver's Heat Map Cluster Analysis of normalized MMP1, 3, 9, 10, 11 and 13 in all samples. (B) Comparison of mRNA expression of MMPs in the three groups, as noted on the x-axis, based on BWH tumor stage and tumor differentiation. Mod = moderate differentiation; Poor = poor differentiation; N = normal; T2, T3 = BWH tumor stage. * and ** represent p<0.05. MMP3, 10, 11 and 13 expression were significantly increased in poorly differentiated cSCC with PNI compared with moderately differentiated cSCC with PNI. However, MMP 11 levels were not different compared with poorly differentiated cSCC with PNI and normal skin tissues (p = 0.325).

recurrence and metastasis [23–26]. Historically, post-operative radiation therapy (PORT) was used for management of perineural SCC [27]; however, the use of PORT with or without a PD-1 inhibitor is being evaluated in a clinical trial (NCT03969004—Study of Adjuvant Cemiplimab Versus Placebo After Surgery and Radiation Therapy in Patients With High Risk Cutaneous Squamous Cell Carcinoma). In contrast, some studies reported no significant changes in outcomes for patients with SCC and PNI. [28, 29] In our study, 3 of 9 patients with cSCC with PNI had poor outcomes eventuating in metastasis and death from disease. On histological examination, the 3 poor outcome patients had BWH T3, poorly differentiated cSCC. In contrast, the other 6 patients had T2A or T2B cSCC with moderate differentiation. Interestingly, all three poor outcome patients exhibited significantly high levels of MAGE-A3 expression in their primary cSCC tumors, whereas the other 6 patients showed no to low MAGE-A3 expression. Thus, MAGE-A3 may be prognostic biomarker for poor outcomes in patients with cSCC with PNI.

MAGE-A4 expression also correlates with poor prognosis of various cancers, and this CTA has been studied as a potential biomarker [30–32]. MAGE-A3 and MAGE-A4 are generally

expressed at similar levels. Brisam M. et al. reported that MAGE-A3 and MAGE-A4 along with -A1, -A5, -A9 and -A11 were significantly correlated with clinically advanced stages of disease based on the analysis of MAGE-A1 to -A12 expression in oral squamous cell carcinoma [33]. These authors also suggested that MAGE-A3 and MAGE-A4 could be used as prognostic markers to improve patient follow-up care. In our study, although MAGE-A4 is over-expressed in poorly differentiated stage 3 cSCC with PNI, the greater increase in MAGE-A3 expression makes MAGE-A3 a better biomarker (average 82- vs 37-fold increase, respectively, compared with normal skin).

Others have demonstrated that MAGE-A3 promotes proliferation and growth of cancer cells through interaction with the tumor suppressor gene p53 and enzyme E3 ubiquitin ligase [34–36]. In our study, p53 mRNA levels do not significantly differ among poorly differentiated cSCC with PNI, moderately differentiated cSCC with PNI and normal skin tissues (347.93 ±69.7 vs. 373.37±47.06 vs. 395.14±66.38; p > 0.05). This finding is consistent with previous studies given that MAGE-A3 is believed to inhibit p53 functions by blocking its interaction with chromatin and accelerating its denaturation and not at the transcription level [35, 36].

We investigated cyclin D, E, A, and B expression to determine any associations with MAGE-A3 expression in stage 3 poorly differentiated cSCC with PNI. Cyclin A, B and E expression was significantly increased compared with normal tissues, which is consistent with the fact that advanced staged tumors exhibit more rapid cell proliferation compared with nor-mal epithelial cells. However, cyclin A is the only cyclin that exhibits a statistical difference between poorly differentiated and moderately differentiated cSCC with PNI (p = 0.0277). In contrast to other studies showing overexpression of cyclin D1 in SCC, we found no significant differences in cyclin D1 expression [37–39]. One possible explanation could be due to the fact that A431 and Pam 212 tumors potentially exhibit overexpression of different isoforms of cyclin D [40]. In our study, cyclin D2 and D3 but not D1 exhibited statistically increased levels in poorly differentiated stage 3 cSCC with PNI compared with normal epithelial tissues (p = 0.0009 and p = 0.0001, respectively). Cyclin D2 even exhibited significant increased expression between poorly differentiated and moderately differentiated cSCC with PNI (p = 0.006).

We postulated that MAGE-A3 might contribute to poor outcomes in patients with cSCC and PNI given its association with tumor proliferation. Prior studies have demonstrated the association of poor outcome with advanced T stage and poor differentiation. In our in vitro experiments with A431 cSCC cells, blockage of MAGE-A3 expression reduced the percentage of S-phase cells and reduced scratch closure after 72 hours. These findings are consistent with previous studies demonstrating that MAGE-A3 modulates cell proliferation [12, 41]. However, cyclin D1 protein levels did not differ among the untreated, ATS alone and pre-treated MAGE-A3 antibody groups. This finding indicates that MAGE-A3 did not modulate the cell cycle via cyclin D1 and might further support the analysis of insignificant changes in cyclin D1 expression in our human tumor samples.

MAGE-A3 protein is located in the endoplasmic reticulum [42]. DMSO can induce water pores in dipalmitoylphosphatidylcholine bilayers, subsequently increasing the permeability of cell membranes and facilitating the entry of antibodies into cells [43]. MAGE-A3 antibodies thus enter cells and interact with MAGE-A3 proteins in the endoplasmic reticulum. Doyle et al. reported that multiple MAGE family proteins are capable of binding E3 RING ubiquitin ligases to form MAGE-RING protein complexes [44]. These complexes promote degradation of p53 in the ubiquitin-protease system, and greater than 50 complexes have been identified [45]. MAGE-A3 proteins specifically interact with TRIM28 E3 RING ubiquitin ligase to form a stable MAGE-RING ligase complex [46, 47]. In addition, previous studies have reported that this complex also accelerates degradation of two important tumor suppressors, fructose-

1,6-bisphosphatase (FBP1) and 5' adenosine monophosphate-activated protein kinase (AMPK) [48, 49]. Altogether, MAGE-A3 downregulates apoptosis and drives tumorigenesis, cell cycle progression and metastasis. Based on these findings, we hypothesize that the decrease in the percentage of cells in S-phase and deceleration of cancer migration observed in our in vitro experiments might be attributed to the blockage of interactions between MAGE-A3 and TRIM E3 RING ubiquitin ligase and disruption of MAGE-RING ligase complex formation by MAGE-A3 antibodies. The upregulation of p53 expression in A431 cells pre-treated with MAGE-A3 antibodies shown in Fig 3B further supports our hypothesis.

To evaluate the role of MAGE-A3 in vivo, we injected Pam 212 murine cSCC cells intradermally into Balb/c mice and measured the expression of MAGE-A3 and cyclins. Six weeks post-injection, tumor volume increased, and MAGE-A3 and cyclin A2, B1 and E1 protein levels were elevated in syngeneic tumors compared to normal murine keratinocytes and Pam 212 cells grown in vitro. This result supports our hypothesis that MAGE-A3 expression is increased in proliferating tumors. We found that Pam 212 cells with MAGE-A3 knockout failed to grow in Balb/c mice, and total regression was observed in two cases. In addition, immunohistochemical analysis revealed down regulation of cyclin B, D and E expression in MAGE-A3 knockout tumors. This finding suggests that the reduction of MAGE-A3 expression leads to changes in cyclins and results in tumor regression.

Interestingly, our experiments also indicate that MAGE-A3 is more highly expressed in Pam 212 cells grown in vivo compared with those grown in vitro. We believe this upregulation observed is associated with ECM changes in mice. ECM plays an important role in cancer progression. Lu et al. demonstrated that tumors display desmoplasia, which can alter the organization and enhancing post-translational modifications of ECM proteins [50]. Others reported that cell-to-cell adhesive signaling in the ECM can alter tumor behavior [51–53]. Nanostring analysis of our human cSCC samples revealed no statistical difference in collagen I, II, III, IV, V and VI levels between poorly differentiated or moderately differentiated PNI-cSCC and normal skin tissues. However, levels of COL11A1, which encodes collagen XI alpha 1 chain, are significantly increased when cSCC advances from moderate to poor differentiation (1132.56 ±882.7 vs. 197.6±267.2; p = 0.014). Similar findings were also reported in the study of Li et al. in esophageal SCC patients; however, we did not find any changes in other collagens [54]. Increased collagen XI expression is associated with fibroblasts in various cancers, including lung and breast, and is considered as a potential biomarker for cancer invasiveness [55–57]. Sok J.C. et al. revealed that knockout of collagen XI alpha I gene in a head and neck SCC cell line results in significant reductions in cancer proliferation, invasion and migration [55].

In addition, fibronectin (FN), a large heterodimeric glycoprotein, is a well-studied ECM protein in cancer progression. Increased FN levels are associated with metastasis, proliferation and poor prognosis in various cancers [58–62]. Liu et al. further demonstrated that down regulation of FN increased MAGE-A3 expression, which further promoted cell migration and invasion of thyroid carcinoma in vitro and lung metastasis in vivo by inhibiting p53 functions [13]. However, we did not find any significant changes in FN1 expression between moderately and poorly differentiated cSCC tumors with PNI and normal skin tissues (3023.8±1996.4 vs. 1432 ±1487.9 vs. 1686.5±2135.4; p>0.05). A larger sample size may help further clarify the correlation among collagens, FN and tumor growth. On the other hand, MMPs have long been known to facilitate cancer progression, cell invasion and metastasis through degradation of the ECM [63–65]. In our study, MMP3, 10 and 13 mRNA expression in poorly differentiated cSCC with PNI is significantly increased compared with moderately differentiated cSCC with PNI and normal skin tissues (p<0.05). These changes indicate that the interactions between cSCC with PNI and ECM are greatly increased as the cancer progresses from moderate differentiation to poor differentiation.

In summary, we found that cSCC tumors with PNI tend to exhibit poor clinical outcomes in the context of increased MAGE-A3 expression, advanced tumor stage and poor differentiation. Both in vitro and in vivo models suggested that upregulation of MAGE-A3 promotes tumor progression via modulation of cyclin levels. Taken together, this study highlights the prognostic value of MAGE-A3 in patients with cSCC with PNI via acceleration of cell proliferation.

## Supporting information

**S1 Table. Clinical characteristics of patients with cSCC and PNI.**
(DOCX)

**S1 Raw images. Original images of blots and gels reported in the manuscript.**
(PDF)

## Author Contributions

**Conceptualization:** Nicole Doudican, Diane Felsen, John A. Carucci.

**Data curation:** Aaron Chen, Alexis L. Santana, Nicole Doudican, Nazanin Roudiani.

**Formal analysis:** Aaron Chen, Alexis L. Santana, Nicole Doudican, Jean-Philippe Therrien, James Lee, John A. Carucci.

**Funding acquisition:** John A. Carucci.

**Investigation:** Alexis L. Santana, Nicole Doudican, Nazanin Roudiani, Kristian Laursen.

**Methodology:** Alexis L. Santana, Nicole Doudican, Nazanin Roudiani, Kristian Laursen, Jean-Philippe Therrien, James Lee, Diane Felsen.

**Project administration:** Nicole Doudican, John A. Carucci.

**Resources:** Jean-Philippe Therrien, James Lee, John A. Carucci.

**Software:** Aaron Chen, Nicole Doudican.

**Supervision:** John A. Carucci.

**Validation:** Nicole Doudican, John A. Carucci.

**Visualization:** Aaron Chen, Nicole Doudican, John A. Carucci.

**Writing – original draft:** Aaron Chen, Alexis L. Santana, Nicole Doudican, Nazanin Roudiani, Kristian Laursen, Jean-Philippe Therrien, James Lee, Diane Felsen, John A. Carucci.

**Writing – review & editing:** Aaron Chen, Nicole Doudican, John A. Carucci.

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
