## [Decision Letter · Decision Letter 0]

26 Feb 2020

PONE-D-19-34576

MAGE-A3 is a prognostic biomarker for poor clinical outcome in cutaneous squamous cell carcinoma with perineural invasion via modulation of cell proliferation

PLOS ONE

Dear Dr. Carucci,

Thank you for submitting your manuscript to PLOS ONE. After careful consideration, we feel that it has merit but does not fully meet PLOS ONE’s publication criteria as it currently stands. Therefore, we invite you to submit a revised version of the manuscript that addresses the points raised during the review process.

We would appreciate receiving your revised manuscript by Apr 11 2020 11:59PM. To enhance the reproducibility of your results, we recommend that if applicable you deposit your laboratory protocols in protocols.io, where a protocol can be assigned its own identifier (DOI) such that it can be cited independently in the future. For instructions see: http://journals.plos.org/plosone/s/submission-guidelines#loc-laboratory-protocols

We look forward to receiving your revised manuscript.

Kind regards,

Yoshihiko Hirohashi, M. D., Ph. D.

Academic Editor

PLOS ONE

Additional Editor Comments (if provided):

Dear Dr. John A. Carucci,

As pointed out by reviewer's, please re-consider about specific concerns. Especially, what is the mechanisms that anti-MAGE-A3 antibody have functions on A431 cells. Is MAGE-A3 protein expressed on cell surface?

Sincerely yours,

Yoshihiko Hirohashi, M. D., Ph. D.

Academic Editor

PLOS ONE

Journal Requirements:

2. At this time, we request that you  please report additional details in your Methods section regarding animal care, as per our editorial guidelines:

(1) Please state the number of mice used in the study  

(2) Please provide details of animal welfare (e.g., shelter, food, water, environmental enrichment)

(3) Please describe any steps taken to minimize animal suffering and distress, such as by administering anaesthesia  

(4) Please include the method of euthanasia

(5) Please provide the formula used to calculate tumor volume  

(6) Please describe the post-operative care received by the animals, including the frequency of monitoring and the criteria used to assess animal health and well-being.

Thank you for your attention to these requests.

     Jean-Philippe Therrien and James Lee were employees and shareholders of GlaxoSmithKline while these studies were being performed. 

Reviewers' comments:

Reviewer's Responses to Questions

**Comments to the Author**

1. Is the manuscript technically sound, and do the data support the conclusions?

Reviewer #1: No

Reviewer #2: Yes

2. Has the statistical analysis been performed appropriately and rigorously? 

Reviewer #1: Yes

Reviewer #2: N/A

3. Have the authors made all data underlying the findings in their manuscript fully available?

Reviewer #1: Yes

Reviewer #2: Yes

4. Is the manuscript presented in an intelligible fashion and written in standard English?

Reviewer #1: Yes

Reviewer #2: Yes

5. Review Comments to the Author

Reviewer #1: In this paper, the authors examined the expression of MAGE-A3, cyclin proteins, collagens, and MMPs in cutaneous SCC (cSCC) with perineural invasion (PNI). They found that MAGE-A3 and some of cyclins, collagens, and MMPs were upregulated in poorly differentiated cSCC with PNI. Furthermore, they also investigated the role of MAGE-A3 in cancer progression using in vivo and in vitro models. Although this manuscript contains some interesting findings, there are several important issues that need to be improved before publication.

Major points:

1) Cutaneous SCC is the common cancer and is not rare. Thus, the authors should be able to collect more samples. The number of patients with cSCC with PNI (only 9 patients) is too small to reach a conclusion.

2) The authors must describe the details of the clinical characteristics of patients with cSCC with PNI.

3) The authors showed that not only MAGE-A3 but also MAGE-A4 mRNA expression was upregulated in poorly differentiated cSCC with PNI in Figure 1A. How about the expression of MAGE-A4 protein in tissue from the patients with cSCC with PNI? The authors should also examine the MAGE-A4 expression in tissue using immunohistochemistry.

4) What is the minimum percentage of MAGE-A3 positive tumor cells required to consider the sample MAGE-A3 positive?

5) Representative immunohistological futures of not only poorly and moderately differentiated cSCC but also healthy controls should be shown in Figure 1B.

6) In Figure 2, the authors treated A431 SCC cells with anti-MAGE-A3 antibody to explore the role of MAGE-A3 in cell cycle progression. What is the mechanism in this experiment? Does MAGE-A3 express on cell surface of A431 SCC cells? Furthermore, the authors compared A431 SCC cells treated with MAGE-A3 antibody with those treated with vehicle or untreated. However, A431 SCC cells treated with isotype control antibody should be used as a control group.

If the authors want to examine the role of MAGE-A3 in A431 SCC cells, it may be desirable to knockdown or knockout expression of MAGE-A3 using siRNA or CRISPR/Cas9.

7) In page 15, the authors described that the trend of increased MAGE-A3 expression in Pam 212 SCC tumors was confirmed by immunohistochemical staining. However, there are no immunohistological images.

8) In Figure 4, knockout of MAGE-A3 expression via CRISPR/Cas9 was confirmed by PCR and qPCR. However, the authors should also perform western blot analysis to prove knockout of MAGE-A3 protein.

Miner points:

1) Please describe the details of MAGE-A3 antibody such as company, clone name, and dilution rate in EDU assays and Immunohistochemistry of the materials and methods section (page 9 and 10).

Reviewer #2: In “MAGE-A3 is a prognostic biomarker for poor clinical outcome in cutaneous squamous cell carcinoma with perineural invasion via modulation of cell proliferation,” Chen et al., demonstrate the prognostic value of MAGE-A3 in clinical outcomes of patients with cutaneous squamous cell carcinoma with perineural invasion. They go on elucidate mechanisms by which MAGE-A3 can positively influence tumor migration and growth. Overall, the study makes the novel association between a cancer-associated gene and squamous cell carcinoma, a skin cancer with significant morbidity and mortality and few specific molecular targets. Further, the authors establish mechanisms by which this gene and protein product impacts cancer cell behavior.

SPECIFIC COMMENTS

Figure 2, Please report whether MAGE-A3 expressed in A431 cells in vitro by mRNA or protein.

Figure 2A, Please include results from IgG1 isotype control treatment if such reagent exists and if such experiments were performed. In the legend, “A341” should be “A431.” Further, please provide any historical data or cited literature indicating that the cellular localization of MAGE-A3 has access to antibody on the surface of cells.

Figure 2B, Please include statistical significance data on the graph.

Figure 2D, Please describe the use of the anti-p53 Western Blot antibody used in the methods

Figure 3A, Please include the number of mice used per cohort & how many experiments were conducted. Further, in the text, the description of the change in growth is reported without units (1112.24?...) – please include units. This is also the case later in the text – please include units numerical values throughout.

Figure 3B, Cyclin D2/D3 does not appear to be elevated in PAM212 tumors as text suggests. Quantification not provided. Please clarify this statement.

“The trend of increased MAGE-A3 expression in Pam 212 SCC tumors was further confirmed by immunohistochemical staining (Figure 3C)” on page 21 –while Figure 3C reports relative RNA expression. Please clarify this statement.

MTS proliferation is noted in the methods but not includes in the Figures. Please remove from the methods if this data is not to be included.

Figure 4B, Please include the statistical details in the graph

Figure 4C, Please report the number of mice per cohort & how many experiments were performed; Please include the statistical significance in the graph.

Figure 4D & 4E are appear to be transposed compared to what is described in the text on page 16.

6. PLOS authors have the option to publish the peer review history of their article (what does this mean?). If published, this will include your full peer review and any attached files.

Reviewer #1: No

Reviewer #2: No

---

## [Author Response · Author response to Decision Letter 0]

25 Sep 2020

Editor Comments

Major comments: 

I. What is the mechanisms that anti-MAGE-A3 antibody have functions on A431 cells. Is MAGE-A3 protein expressed on cell surface?

We thank the editor for raising this important question. MAGE-A3 protein is localized to the endoplasmic reticulum. (Morishima, Nakanishi et al. J Biol Chem 2002 277(37): 34287-34294) Doyle et al discovered that multiple MAGE family proteins are capable of binding E3 RING ubiquitin ligases to form MAGE-RING protein complexes. (Doyle, Gao et al. Mol Cell 2010 39(6): 963-974) These complexes promote degradation of p53 in the ubiquitin-protease system, and more than 50 complexes have been identified. (Lee and Potts. J Mol Biol 2017 429(8): 1114-1142) MAGE-A3 protein specifically interacts with TRIM28 E3 RING ubiquitin ligase to form a stable MAGE-RING ligase complex. (Yang, O’Herrin et al. Cancer Res 2007 67(20): 9954-9962; Pineda and Potts. Autophagy 2015 11(5): 844-846) In addition, previous studies have found this complex also accelerates degradation of two important tumor suppressors, fructoses-1,6-bisphosphatase (FBP1) and 5’ adenosine monophosphate-activated protein kinases (AMPK). (Jin, Pan et al. Oncogenesis 2017 6(4): e312; Ye, Xie et al. Cell Physiol Biochem 2018 45(3): 1205-1218) Together, MAGE-A3 downregulates apoptosis and drives tumorigenesis, cell cycle progression and metastasis. Based on these findings, we hypothesize that the decrease in the percentage of cells in S-phase and deceleration of cancer migration observed in our in vitro experiments might be attributed to the blockage of interactions between MAGE-A3 and TRIM E3 RING ubiquitin ligase and disruption of MAGE-RING ligase complex formation by MAGE-A3 antibodies. The upregulation of p53 expression in A431 cells pre-treated with MAGE-A3 antibodies shown in Figure 3B further supports our hypothesis. (Page 24 Line 9 to Page 25 Line 3) We prepared an antibody transport solution (ATS) that contained PBS, 50% glycerol, 0.02% sodium azide (pH 7.3) and 0.1% dimethyl sulfoxide (DMSO) to assist MAGE-A3 antibodies to enter A431 cells.(Page 10 Line 18-20) DMSO can induce water pores in dipalmitoylphosphatidylcholine bilayers and cause cell membrane to become floppier, which increases permeability for the antibodies to enter cells. (Notman, Noro et al. J Am Chem Soc 128(43): 13982-13983) (Page 24 Line 9-11) 

II. PLOS ONE now requires that authors provide the original uncropped and unadjusted images underlying all blot or gel results reported in a submission’s figures or Supporting Information files.

We thank the editor for this important reminder. The original uncropped and unadjusted images of all blot or gel results reported are included in the supporting file “S1_raw_images.pdf”.

Additional comments:

(1) Please state the number of mice used in the study

We thank the editor for this comment. PAM 212 was injected intradermally into 5 balb/c mice, and tumor growth and MAGE-A3 and cyclin expression were assessed. (Page 18 Line 12) In the in vivo MAGE-A3 KO tumor growth study, 3 mice were included in each group and the experiment was conducted twice for a total of 18 mice. (Page 19 Line 20)

(2) Please provide details of animal welfare (e.g., shelter, food, water, environmental enrichment)

We thank the editor for this suggestion. This information has been added to the Materials and Methods section. Mice were maintained in ventilated cages and fed/watered ad libitum with experiments carried out under an IACUC approved protocol (160103-01) as well as following institutional guidelines for the proper and humane use of animals in research. (Page 7 Line 14-17)

(3) Please describe any steps taken to minimize animal suffering and distress, such as by administering anaesthesia. 

We thank the editor for this comment. This information has been added to the Materials and Methods section. For tumor growth experiments, 5.0 x 106 Pam 212 or Pam 212 guide RNA (520 or 521) cells were injected intradermally with Matrigel under anesthesia to minimize pain and discomfort. (Page 7 Line 17-19)

(4) Please include the method of euthanasia

We thank the editor for this comment. This information has been added to the Materials and Methods section. Mice were sacrificed by CO2 euthanasia. (Page 7 Line 21)

(5) Please provide the formula used to calculate tumor volume 

We thank the editor for this comment. This information has been added to the Materials and Methods section. Tumor volume was calculated using the formula a 2 �b/2, in which a is the short diameter of the tumor and b is the long diameter of the tumor. (Page 8 Lines 2-4)

(6) Please describe the post-operative care received by the animals, including the frequency of monitoring and the criteria used to assess animal health and well-being.

We thank the editor for helping clarify our animal control protocols. This information has been added to the Materials and Methods section. After tumor implantation, mice were monitored thrice weekly for signs of pain or distress during the course of the experiment. Mice meeting any of the following criteria were immediately euthanized via by CO2 euthanasia: ulcerated tumors, mice that experience greater that 20% of weight loss, mice that show signs of difficulty breathing or appear moribund, mice that experience difficulties moving freely in the cage, or tumor burden greater than 15% body weight. (Page 7 Line 19 to Page 8 Line 2)

Reviewer #1 

Major points:

1) Cutaneous SCC is the common cancer and is not rare. Thus, the authors should be able to collect more samples. The number of patients with cSCC with PNI (only 9 patients) is too small to reach a conclusion.

The reviewer has raised an excellent question. Cutaneous SCC is the second most common human cancer. However, PNI is diagnosed mainly by incidental pathological finding due to the fact that most patients with cSCC and PNI have no clinical symptoms and no radiologic evidence of PNI. Its reported incidence rates range between ~2 and 5 %. For this study, we evaluated 24 patients with cSCC, and 9 exhibited PNI (37.5% in this cohort). We were limited with respect to total number of patients evaluated by budgetary concerns. Results of this limited series showed upregulation of MAGE-A3 mRNA expression in the patients showing poor prognosis and we investigated possible mechanisms driving this correlation. Utility of MAGE as a biomarker for prognosis is currently under study by our group. 

2) The authors must describe the details of the clinical characteristics of patients with cSCC with PNI.

We thank the reviewer for this important suggestion. The clinical characteristics of patients with cSCC with PNI is provided in the supporting file “S1 table.docx” and presented below.

Patient

No. Age Gender Tumor Site Tumor Size (cm) IHC Score Differentiation BWH Stage

1 86 M Scalp 2 < 20 Mod 2A

2 96 F Forehead 2.5 20 Mod 2A

3 81 M Face 2.5 < 20 Mod 2B

4 81 M Temple 1.4 < 20 Mod 2B

5 64 M Temple 3.1 21-49 Mod 2B

6 45 F Forehead 2.5 21-49 Mod 2B

7 65 M Ear 2.5 50-999 Poor 3

8 90 M Arm 2 >1000 Poor 3

9 88 M Scalp 2 >1000 Poor 3

3) The authors showed that not only MAGE-A3 but also MAGE-A4 mRNA expression was upregulated in poorly differentiated cSCC with PNI in Figure 1A. How about the expression of MAGE-A4 protein in tissue from the patients with cSCC with PNI? The authors should also examine the MAGE-A4 expression in tissue using immunohistochemistry.

We thank the reviewer for this important suggestion. We have split Figure 1 into Figures 1 and 2. IHC assessment of MAGE-A4 expression in normal skin, moderately differentiated cSCC and poorly differentiated cSCC with PNI have been added to Figure 2, and this information has been added to the text on page 15 line 15.

4) What is the minimum percentage of MAGE-A3 positive tumor cells required to consider the sample MAGE-A3 positive?

We thank the reviewer for this question about a key measurement in our Nanostring technology analysis. This information has been added to the Material and Methods section. MAGEA3 positivity is defined by a normalized NanoString value of greater than 20 as described in our previous study (Abikhair et al. J Invest Dermatol. 2017 Mar;137(3):775-778). (Page 6 Line 14-15) 

5) Representative immunohistological futures of not only poorly and moderately differentiated cSCC but also healthy controls are shown in Figure 1B.

We thank the reviewer for this important suggestion. Figure 1B has been modified and changed to Figure 2. The immunohistological features of healthy control has been added into Figure 2 for both MAGE-A3 and MAGE-A4, and this information has been added to the text on page 15 line 15.

6) In Figure 2, the authors treated A431 SCC cells with anti-MAGE-A3 antibody to explore the role of MAGE-A3 in cell cycle progression. What is the mechanism in this experiment? Does MAGE-A3 express on cell surface of A431 SCC cells? Furthermore, the authors compared A431 SCC cells treated with MAGE-A3 antibody with those treated with vehicle or untreated. However, A431 SCC cells treated with isotype control antibody should be used as a control group. If the authors want to examine the role of MAGE-A3 in A431 SCC cells, it may be desirable to knockdown or knockout expression of MAGE-A3 using siRNA or CRISPR/Cas9.

We thank the reviewer for raising this important question. Figure 2 has been changed to Figure 3. MAGE-A3 protein is localized to the endoplasmic reticulum. (Morishima, Nakanishi et al. J Biol Chem 2002 277(37): 34287-34294) Doyle et al discovered that multiple MAGE family proteins are capable of binding E3 RING ubiquitin ligases to form MAGE-RING protein complexes. (Doyle, Gao et al. Mol Cell 2010 39(6): 963-974) These complexes promote degradation of p53 in the ubiquitin-protease system, and more than 50 complexes have been identified. (Lee and Potts. J Mol Biol 2017 429(8): 1114-1142) MAGE-A3 protein specifically interacts with TRIM28 E3 RING ubiquitin ligase to form a stable MAGE-RING ligase complex. (Yang, O’Herrin et al. Cancer Res 2007 67(20): 9954-9962; Pineda and Potts. 2015 Autophagy 11(5): 844-846) In addition, previous studies have found this complex also accelerates degradation of two important tumor suppressors, fructoses-1,6-bisphosphatase (FBP1) and 5’ adenosine monophosphate-activated protein kinases (AMPK). (Jin, Pan et al. Oncogenesis 2017 6(4): e312; Ye, Xie et al. Cell Physiol Biochem 2018 45(3): 1205-1218) Together, MAGE-A3 downregulates apoptosis and drives tumorigenesis, cell cycle progression and metastasis. Based on these findings, we hypothesize that the decrease in the percentage of cells in S-phase and deceleration of cancer migration observed in our in vitro experiments might be attributed to the blockage of interactions between MAGE-A3 and TRIM E3 RING ubiquitin ligase and disruption of MAGE-RING ligase complex formation by MAGE-A3 antibodies. The upregulation of p53 expression in A431 cells pre-treated with MAGE-A3 antibodies shown in Figure 3B further supports our hypothesis. (Page 24 Line 9 to Page 25 Line 3) 

We used actin monoclonal IgG1-kappa antibody (ThermoFisher #MA5-11869) as an isotype control antibody (IgG1, kappa), and this information has been added to the Methods section on page 11 line 17-18. We have conducted the scratch tests using this antibody as an isotype control, and the results have been included into Figure 3C and 3D. This information is presented in the text on page 17 lines 10-15.

Dimethyl sulfoxide (DMSO) can induce water pores in dipalmitoyl phosphatidylcholine bilayers and cause cell membrane to become floppier, which increases permeability for the antibodies to enter cells. (Notman, Noro et al. J Am Chem Soc 128(43): 13982-13983) (Page 24 Line 9-11) Thus, we prepared an antibody transport solution (ATS) that contained PBS, 50% glycerol, 0.02% sodium azide (pH 7.3) and 0.1% dimethyl sulfoxide (DMSO) to assist MAGE-A3 antibodies to enter A431 cells. (Page 10 Lines 18-20) The results demonstrated suppression of cancer proliferation and migration; therefore, we decided to knockout expression of MAGE-A3 by CRISPR/Cas9 in PAM212 and follow-up its syngeneic tumor growth in balb/c mice in 6 weeks. This in vivo experiment reveals that knockout expression of MAGE-A3 results in significant tumor regression.

7) In page 15, the authors described that the trend of increased MAGE-A3 expression in Pam 212 SCC tumors was confirmed by immunohistochemical staining. However, there are no immunohistological images.

We apologize for this error. The sentence has been removed.

8) In Figure 4, knockout of MAGE-A3 expression via CRISPR/Cas9 was confirmed by PCR and qPCR. However, the authors should also perform western blot analysis to prove knockout of MAGE-A3 protein.

We thank the reviewer for this comment. We have performed western blot analysis to show knockout of MAGE-A3 protein expression. The findings have been added to Figure 5C, and this information has been presented on page 19 line 17-19. 

Minor points:

1) Please describe the details of MAGE-A3 antibody such as company, clone name, and dilution rate in EDU assays and Immunohistochemistry of the materials and methods section (page 9 and 10).

We thank the reviewer for this comment. The details of MAGE-A3 antibody has been added on page 11 line 15. Dilution rates in EDU assays and immunohistochemistry have been added on page 10 line 18 and page 12 line 9.

Reviewer #2 

SPECIFIC COMMENTS

1) Figure 2, Please report whether MAGE-A3 expressed in A431 cells in vitro by mRNA or protein.

We thank the reviewer for this comment. MAGE-A3 protein localizes to the endoplasmic reticulum are the targets of MAGE-A3 antibodies in our A431 experiments. (Page 24 Line 9) The original figure 2 has been changed to Figure 3. (Page 17 Line 4)

2) Figure 2A, Please include results from IgG1 isotype control treatment if such reagent exists and if such experiments were performed. In the legend, “A341” should be “A431.” Further, please provide any historical data or cited literature indicating that the cellular localization of MAGE-A3 has access to antibody on the surface of cells.

We thank the reviewer for this comment. Actin monoclonal IgG1-kappa antibody (ThermoFisher #MA5-11869) was used in A431 experiments as an isotype control (IgG1, kappa). (Page 11 Line 17) 

We have corrected the typo and apologize for the error. 

We have provided cited literature and a description as follows: MAGE-A3 protein is located to the endoplasmic reticulum. (Morishima, Nakanishi et al. J Biol Chem 2002 277(37): 34287-34294) (Page 24 Line 9) Dimethyl sulfoxide (DMSO) can induce water pores in dipalmitoyl phosphatidylcholine bilayers and cause cell membrane to become floppier, which increases permeability for the antibodies to enter cells.( Notman, Noro et al. J Am Chem Soc 128(43): 13982-13983) (Page 24 Line 9-11) Thus, we prepared an antibody transport solution (ATS) that contained PBS, 50% glycerol, 0.02% sodium azide (pH 7.3) and 0.1% dimethyl sulfoxide (DMSO) to assist MAGE-A3 antibodies to enter A431 cells. (Page 10 Lines 18-20)

3) Figure 2B, Please include statistical significance data on the graph.

We thank the reviewer for this comment. We have modified Figure 2B with results of isotype antibody into Figure 3C. The legend on page 17 line 10-13 has also changed accordingly. 

4) Figure 2D, Please describe the use of the anti-p53 Western Blot antibody used in the methods

We thank the reviewer for this comment. Figure 2D has been changed to Figure 3B matching to the text on page 17 Line 8. We have added the p53 antibody (Cell Signaling (1C12) Mouse mAb #2524) in the Immunoblot Analysis and Antibodies of the Methods section. (Page 11 Line 16-17) 

5) Figure 3A, Please include the number of mice used per cohort & how many experiments were conducted. Further, in the text, the description of the change in growth is reported without units (1112.24?...) – please include units. This is also the case later in the text – please include units numerical values throughout.

We thank the reviewer for correcting our errors. Figures 3 and 4 have been changed to Figures 4 and 5. We have added a total of 5 mice done in the experiment shown in Figure 4A. (Page 18 Line 12) The knockout experiment was conducted twice and the results of total 18 mice were analyzed in Figure 5D. (Page 19 Line 20) Units also have been added in the text. (Page 17 Line 21)

6) Figure 3B, Cyclin D2/D3 does not appear to be elevated in PAM212 tumors as text suggests. Quantification not provided. Please clarify this statement.

We thank the reviewer for pointing out this error. We have removed the statement.

7) “The trend of increased MAGE-A3 expression in Pam 212 SCC tumors was further confirmed by immunohistochemical staining (Figure 3C)” on page 21 –while Figure 3C reports relative RNA expression. Please clarify this statement.

We apologize for this error. The sentence has been removed. 

8) MTS proliferation is noted in the methods but not includes in the Figures. Please remove from the methods if this data is not to be included.

We apologize for this error. The section has been removed from the Material and Methods section. 

9) Figure 4B, Please include the statistical details in the graph

We thank the reviewer for the comment. Figure 4B has been changed to Figure 5B. The statistical details have been added in the Figure 5B legend as follows: Fold increase of MAGE-A3 expression in wide type Pam 212 was set to 1. Fold changes in MAGE-A3 expression in KO Pam 212 by Guide 521 and 520 were 0.0049 ± 0.00198 and 0.000535 ± 0.000209, respectively. Error bars have been added into the graph to indicate the standard deviation. The error bars are difficult to see in the graph, so we also added the exact values to the figure legend. (Page 19 Line 14-17)

10) Figure 4C, Please report the number of mice per cohort & how many experiments were performed; Please include the statistical significance in the graph.

We thank the reviewer for the comment. The figure 4C has been changed to Figure 5D and this change has been updated in the manuscript. (Page 19 Line 19) MAGE-A3 KO animal experiments were conducted twice, and each time 3 balb/c mice were included in each group for a total of 18 mice. (Page 19 Line 20) Results with statistical significance in the graph are noted with *, and this information has been added to the figure legend. (Page 19 Line 21-22)

11) Figure 4D & 4E appear to be transposed compared to what is described in the text on page 16.

We thank the reviewer for correcting this confusion. Figure 4D has been changed to Figure 5F, and Figure 4F has been changed to Figure 5G. The changes have been updated in the text on page 20 Lines 1-4.

Moreover, we have updated our Competing Interests statement as follows per the editor’s suggestion: Jean-Philippe Therrien and James Lee were employees and shareholders of GlaxoSmithKline while these studies were being performed. This does not alter our adherence to Plos One policies on sharing data and materials.

---

## [Editor Report · Decision Letter 1]

19 Oct 2020

MAGE-A3 is a prognostic biomarker for poor clinical outcome in cutaneous squamous cell carcinoma with perineural invasion via modulation of cell proliferation

PONE-D-19-34576R1

Dear Dr. John A. Carucci,

We’re pleased to inform you that your manuscript has been judged scientifically suitable for publication and will be formally accepted for publication once it meets all outstanding technical requirements.

Kind regards,

Yoshihiko Hirohashi, M. D., Ph. D.

Academic Editor

PLOS ONE

Additional Editor Comments (optional):

The authors addressed concerning points.
---

## [Editor Report · Acceptance letter]

23 Oct 2020

PONE-D-19-34576R1 

MAGE-A3 is a prognostic biomarker for poor clinical outcome in cutaneous squamous cell carcinoma with perineural invasion via modulation of cell proliferation 

Dear Dr. Carucci:

I'm pleased to inform you that your manuscript has been deemed suitable for publication in PLOS ONE. Congratulations! Your manuscript is now with our production department. 

Kind regards, 

on behalf of

Dr. Yoshihiko Hirohashi 

Academic Editor

PLOS ONE